PyDamage: automated ancient damage identification and estimation for contigs in ancient DNA de novo assembly

Borry Maxime borry@shh.mpg.de 1
Hübner Alexander 1 2
Rohrlach Adam B. 3 4
Warinner Christina warinner@fas.harvard.edu 1 2 5
1 Microbiome Sciences Group, Max Planck Institute for the Science of Human History, Department of Archaeogenetics , Jena , Germany
2 Faculty of Biological Sciences, Friedrich-Schiller Universität Jena , Jena , Germany
3 Population Genetics Group, Max Planck Institute for the Science of Human History, Department of Archaeogenetics , Jena , Germany
4 ARC Centre of Excellence for Mathematical and Statistical Frontiers, The University of Adelaide , Adelaide , Australia
5 Department of Anthropology, Harvard University , Cambridge , MA , United States of America
Aramayo Rodolfo
Electronic publication date: 2021 Jul 27
Publication date: 2021
Volume: 9
Electronic Location ID: e11845
Received 2021 Mar 29; Accepted 2021 Jul 1
Copyright: ©2021 Borry et al.
Copyright year: 2021
Copyright holder: Borry et al.
License: This is an open access article distributed under the terms of the Creative Commons Attribution License, which permits unrestricted use, distribution, reproduction and adaptation in any medium and for any purpose provided that it is properly attributed. For attribution, the original author(s), title, publication source (PeerJ) and either DOI or URL of the article must be cited.
License URL: https://creativecommons.org/licenses/by/4.0/

Keywords: metagenomics, aDNA, ancient DNA, assembly, damage, de novo, automated

Funding: DFG, German Research Foundation 390713860 European Research Council (ERC) under the European Union’s Horizon 2020 Research and Innovation Program 771234 –PALEoRIDER Werner Siemens Foundation (“Paleobiotechnology”) Alexander Hübner was funded by the Deutsche Forschungsgemeinschaft (DFG, German Research Foundation) under Germany’s Excellence Strategy (EXC 2051 –Project-ID 390713860). Adam B Rohrlach was funded by the European Research Council (ERC) under the European Union’s Horizon 2020 research and innovation program under grant agreement no. 771234 –PALEoRIDER. Maxime Borry and Christina Warinner were funded by the Werner Siemens Foundation (”Paleobiotechnology”). The funders had no role in study design, data collection and analysis, decision to publish, or preparation of the manuscript.

==============================
DNA de novo assembly can be used to reconstruct longer stretches of DNA (contigs), including genes and even genomes, from short DNA sequencing reads. Applying this technique to metagenomic data derived from archaeological remains, such as paleofeces and dental calculus, we can investigate past microbiome functional diversity that may be absent or underrepresented in the modern microbiome gene catalogue. However, compared to modern samples, ancient samples are often burdened with environmental contamination, resulting in metagenomic datasets that represent mixtures of ancient and modern DNA. The ability to rapidly and reliably establish the authenticity and integrity of ancient samples is essential for ancient DNA studies, and the ability to distinguish between ancient and modern sequences is particularly important for ancient microbiome studies. Characteristic patterns of ancient DNA damage, namely DNA fragmentation and cytosine deamination (observed as C-to-T transitions) are typically used to authenticate ancient samples and sequences, but existing tools for inspecting and filtering aDNA damage either compute it at the read level, which leads to high data loss and lower quality when used in combination with de novo assembly, or require manual inspection, which is impractical for ancient assemblies that typically contain tens to hundreds of thousands of contigs. To address these challenges, we designed PyDamage, a robust, automated approach for aDNA damage estimation and authentication of de novo assembled aDNA. PyDamage uses a likelihood ratio based approach to discriminate between truly ancient contigs and contigs originating from modern contamination. We test PyDamage on both on simulated aDNA data and archaeological paleofeces, and we demonstrate its ability to reliably and automatically identify contigs bearing DNA damage characteristic of aDNA. Coupled with aDNA de novo assembly, Pydamage opens up new doors to explore functional diversity in ancient metagenomic datasets.

Introduction

Ancient DNA (aDNA) is highly fragmented (Orlando et al., 2021; Warinner et al., 2017). Although genomic DNA molecules within a living organism can be millions to hundreds of millions of base pairs (bp) long, postmortem enzymatic and chemical degradation after death quickly reduces DNA to fragment lengths of less than 150 bp, typically with medians less than 75 bp and modes less than 50 bp (Mann et al., 2018; Hansen et al., 2017). Within the field of metagenomics, many approaches require longer stretches of DNA for adequate analysis, a requirement that particularly applies to functional profiling, which often involves in silico translation steps (Seemann, 2014). For example, in our experiments we observed that FragGeneScan (Rho, Tang & Ye, 2010), a tool designed for gene prediction from short read data, failed to predict open-reading frames in any DNA sequences shorter than 60 bp. If applied directly to highly fragmented ancient metagenomic datasets, such data filtering can introduce biases that interfere with functional analyses when preservation is variable across samples or when comparing ancient samples to modern ones.

Because very short (<100 bp) and ultrashort (<50 bp) DNA molecules pose many downstream analytical challenges, there is a long-standing interest in leveraging the approach of de novo assembly to computationally reconstruct longer stretches of DNA for analysis. With de novo assembly, longer contiguous DNA sequences (contigs), and sometimes entire genes or gene clusters, can be reconstructed from individual sequencing reads (Compeau, Pevzner & Tesler, 2011), which can then be optionally binned into metagenome-assembled genomes (MAGs) (Kang et al., 2015). Such contigs are more amenable to functional profiling, and applying this technique to microbial metagenomics datasets derived from archaeological remains, such as paleofeces and dental calculus, has the potential to reveal ancient genes and functional diversity that may be absent or underrepresented in modern microbiomes (Tett et al., 2019; Wibowo et al., 2021; Brealey et al., 2020). However, because ancient samples generally contain a mixture of ancient bacterial DNA and modern bacterial contaminants, it is essential to distinguish, among the thousands of contigs generated by assembly, truly ancient contigs from contigs that may originate from the modern environment, such as the excavation site, storage facility, or other exogenous sources.

In addition to being highly fragmented, aDNA also contains other forms of characteristic molecular decay, namely cytosine deamination (observed as C → T transitions in aDNA datasets) (Dabney, Meyer & Pääbo, 2013), which can be measured and quantified to indicate the authenticity of an ancient sample, or even an individual sequence (Hofreiter et al., 2001; Briggs et al., 2007). However, tools for inspecting and filtering aDNA damage were primarily designed for genomic and not metagenomic applications, and they are largely unsuited or impractical for use in combination with de novo assembly. For example, PMDTools (Skoglund et al., 2014) operates at the read level, and when subsequently combined with de novo assembly leads to higher data loss and lower overall assembly quality. MapDamage (Ginolhac et al., 2011) and DamageProfiler (Neukamm, Peltzer & Nieselt, 2020) are tools that can be applied to assembled contigs, but require manual contig inspection by the user, which is infeasible for de novo assemblies yielding tens to hundreds of thousands of contigs. Other tools, such as Mapdamage (Jónsson et al., 2013), do provide an estimation of damage, but use slower algorithms that do not scale well to the analysis of many thousands of contigs. Even tools, such as HOPS (Hübler et al., 2019), designed for aDNA metagenomics, can not easily scale for the analysis of the sheer number of unknown contigs generated by the assembly process. A faster, automated approach with a better sensitivity for distinguishing truly ancient contigs from modern environmental contigs is needed.

Here, we present PyDamage, a software tool to automate the process of contig damage identification and estimation. PyDamage models aDNA damage from deamination data (C → T transitions), and tests for damage significance using a likelihood ratio test to discriminate between truly ancient contigs and contigs originating from modern contaminants. Testing PyDamage on in silico simulated data, we show that it is able to accurately distinguish ancient and modern contigs. We then apply PyDamage to de novo assembled DNA from ancient paleofeces from the site of Cueva de los Muertos Chiquitos, Mexico (ca. 1300 BP) and find that the contigs PyDamage identifies as ancient are consistent with taxa known to be members of the human gut microbiome. Among the ancient contigs, PyDamage authenticated multiple functional genes of interest, including a multidrug and bile salt resistance gene cluster from the gut microbe Treponema succinifacians, a species that is today only found in societies practicing traditional forms of subsistence. Using PyDamage, de novo assembled contigs from aDNA datasets can be rapidly and robustly authenticated for a variety of downstream metagenomics applications.

Material and Methods

Simulated sequencing data

In order to evaluate the performance of PyDamage with respect to the GC content of the assembled genome, the sequencing depth along the genome, the amount of observed aDNA damage on the DNA fragments, and the mean length of these DNA fragments, we simulated short-read sequencing data using gargammel (Renaud et al., 2017) varying these four parameters. We chose three microbiome-associated microbial taxa with low(Methanobrevibacter smithii, 31%), medium (Tannerella forsythia, 47%), and high (Actinomyces dentalis, 72%) GC content, following Mann et al. (2018) (Fig. 1A). Using three different read length distributions (Fig. 1B), we generated short-read sequencing data from each reference genome using gargammel’s fragSim. To the resulting short-read sequences we added different amounts of aDNA damage using gargammel’s deamSim so that ten levels of damage ranging from 0% to 20% were observed, which were measured as the amount of observed C → T substitutions on the terminal base at the 5′ end of the DNA fragments (Fig. 1C). Finally, each of these 90 simulated datasets was subsampled to generate nine coverage bins ranging from 1-fold to 500-fold genome coverage by randomly drawing a coverage value from the uniform distribution defining each bin (Fig. 1D) and these were aligned to their respective reference genome using BWA aln (Li & Durbin, 2009) with the non-default parameters optimized for aDNA -n 0.01 -o 2 -l 16500 (Meyer et al., 2012).

Figure 1 Simulation scheme for evaluating the performance of PyDamage.

(A) The GC content of the three microbial reference genomes. (B) The read length distributions used as input into gargammel fragSim. (C) The amount of aDNA damage as observed as the frequency of C → T substitutions on the terminal 5′ end of the DNA fragments that was added using gargammel deamSim. (D) Nine coverage bins from which the exact coverage was sampled by randomly drawing a number from the uniform distribution defining the bin. (E) Nine contig length bins from which the exact contig length was sampled by randomly drawing a number from the uniform distribution defining the bin.

Test contigs of different length were simulated by defining nine contig length bins ranging from 0.5 kb to 500 kb length (Fig. 1E) and randomly drawing 100 contig lengths from the respective uniform distribution defining each bin. Next, we chose the location of these test contigs by randomly selecting a contig from all contigs of sufficient length. We determined the exact location on the selected test contig from the reference genome by randomly drawing the start position from the uniform distribution defined by the length of the selected reference contig. This resulted in 900 test contigs per reference genome. Using these test contigs, we selected the aligned DNA fragments of the simulated sequencing data that overlapped the region defined by the contig and evaluated them using PyDamage. In total, we evaluated 702,900 test contigs (243,000 contigs for both M. smithii and T. forsythia, and 216,000 contigs for A. dentalis, for which no reference contig longer than 200 kb was available).

Archaeological sample

Preparation and sequencing

We re-analyzed ancient metagenomic data from the archaeological paleofeces sample ZSM028 (Zape 28) dating to ca. 1300 BP from the site of Cueva de los Muertos Chiquitos, in Mexico, previously published in Borry et al.(2020) (ENA run accession codes ERR3678595, ERR3678598, ERR3678602, ERR3678603, and ERR3678613).

Bioinformatic processing

The ZSM028 sample was first trimmed to remove adapters, low quality sequences with Q-scores below 20, and short sequences below 30 bp using AdapterRemoval (Schubert, Lindgreen & Orlando, 2016) v2.3.1. The reads were de novo assembled into contigs using MetaSPAdes Nurk et al. (2017) v3.13.1 using the non-default k-mer lengths 21, 33, and 45. The set of k-mer lengths was adapted to consider the on-average short length of the DNA molecules of this sample (mode: 37 bp). We selected a k-mer length of 45 as the longest one since this was the next longer uneven k-mer length of the median DNA molecule length (median: 44 bp). Reads were then mapped back to the contigs with length > 1,000 bp using Bowtie2 (Langmead & Salzberg, 2012), in the very-sensitive mode, while allowing up to 1 mismatch in the seeding process. The alignment files were then given as an input to PyDamage v0.50. Contigs passing filtering thresholds were functionally annotated with Prokka v1.14.6 (Seemann, 2014), using the –metagenome flag.

Contig Taxonomic Profiling

To investigate the taxonomic profile of the contigs that passed the PyDamage filtering, we ran Kraken2 v2.1.1 (Wood, Lu & Langmead, 2019) using the PlusPFP database( https://benlangmead.github.io/aws-indexes/k2) from 27/1/2021. We then generated the Sankey plot using Pavian (Breitwieser & Salzberg, 2016).

PyDamage implementation

PyDamage takes alignment files of reads (in SAM, BAM, or CRAM format) mapped against reference sequences (i.e., contigs, a MAG, a genome, or any other reference sequences of DNA). For each read mapping to each reference sequence j, using pysam (pysam developers, 2018), we count the number of apparent C → T transitions at each position which is i bases from the 5′ terminal end, i ∈ {0, 1, …, k}, denoted Nij (by default, we set k = 35). Similarly we denote the number of observed conserved ‘C-to-C’ sites Mij, thus Mj=M0j,…,MkjandNj=N0j,…,Nkj.

Finally, we calculate the proportion of C → T transitions occurring at each position, denoted pij, in the following way: p ˆij=NijMij+Nij.

For Di, the event that we observe a C → T transition i bases from the terminal end, we define two models: a null model M0 (Eq. (1)) which assumes that damage is independent of the position from the 5′ terminal end, and a damage model M1 (Eq. (2)) which assumes a decreasing probability of damage the further a the position from the 5′ terminal end. For the damage model, we re-scale the curve to the interval defined by parameters dpminj,dpmaxj. (1) P0Dip0,j=p0=M0πj

(2) P1Dipdj,dpminj,dpmaxj,j=1−pdji×pdj−p ˆminjp ˆmaxj−p ˆminj×dpmaxj−dpminj+dpminj=M1πij,

where p ˆminjpjd=1−pdjk×pdjandp ˆmaxjpjd=1−pdj0×pdj.

Using the curve fitting function of Scipy (Virtanen et al., 2020), with a trf (Branch, Coleman & Li, 1999) optimization and a Huber loss (Huber, 1992), we optimize the parameters of both models using pij, by minimising the sum of squares, giving us the optimized set of parameters θ ˆ0=p ˆ0andθ ˆ1=p ˆdj,d ˆjpmin,d ˆjpmax

for M0 and M1 respectively. Under M0 and M1 we have the following likelihood functions L0θ ˆ0 |Mj,Nj= ∏i=0kMij+NijNijM0π ˆjNij1−M0π ˆjMij,

L1θ ˆ1Mj,Nj= ∏i=0kMij+NijNijM1π ˆijNij1−M1π ˆi1,jMij,

where M0π ˆj and M1π ˆij are calculated using Eqs. (1) and (2). Note that if dpmaxj=dpminj=p0, then M0πj=M1πij for i = 0, …, k. Hence to compare the goodness-of-fit for models M0 and M1 for each reference, we calculate a likelihood-ratio test-statistic of the form λj=−2lnL0θ ˆ0Mj,NjL1θ ˆ1Mj,Nj,

from which we compute a p-value using the fact that λj∼χ22, asymptotically (Neyman & Pearson, 1933). Finally, we adjust the p-values for multiple testing of all references, using the StatsModels (Seabold & Perktold, 2010) implementation of the Benjamini–Hochberg procedure (Benjamini & Hochberg, 1995).

Results

Statistical analysis and model selection

To test the performance of PyDamage in recognizing metagenome-assembled contigs with ancient DNA damage, we used the simulated short-read sequencing data aligned against simulated contigs of different lengths. Our method correctly identified contigs as not significantly damaged for simulations with no damage in 100% of cases. However, our model only correctly identified contigs as significantly damaged in 87.71% of cases where the contigs were simulated to have damage. To assess the performance of our method, and to determine the simulation parameters that most affected model accuracy, we analysed the simulated data using logistic regression via the glm function as implemented in the stats package using R (R Core Team, 2018). We included as potential explanatory variables the median read length, the simulated coverage, the simulated contig length, the simulated level of damage, and the GC content of each of the reference contigs, yielding 32 candidate logistic regression models.

We separated the data into two data sets: half of our data was used as ‘fit data’, data for performing model fit and parameter estimation, and the remaining half was reserved as ‘test data’, data that is used to assess model accuracy on data not used in fitting the model (n = 206, 831 in both cases). Unfortunately, with so many observations in our model, classical model selection methods such as AIC and ANOVA tend to overfit (Babyak, 2004). Similarly, we also performed ten-fold down-sampling of the data for each model such that we had equal numbers of damaged and undamaged simulations so as not to bias the predictive model. Hence, for each of the fitted 32 logistic regression models (with ϵ = 1 × 10−14 and maximum iterations 103) we instead report the mean F1 and Nagelkerke’s R2 values for each candidate model.

Of the 32 candidate models, four models had both F1 and R2 values greater than 0.6 (see Table 1). Each these four models contained at least the following predictor variables: contig length, mean coverage, and the simulated level of damage. However, the full model with GC content and read length as additional predictor variables had similar F1 and R2 values, and so we consider all four models (see Fig. 2). Because it is possible that there is correlation between some of our predictor variables (i.e., increased levels of simulated damage could lead to a reduced median read length), we then performed a Relative Weights Analysis (RWA) to further estimate predictor variable importance in an uncorrelated setting (Chan, 2020). In essence, RWA calculates the proportion of the overall R2 for the model that can be attributed to each variable. We performed RWA on both the full model and our best performing model. We found that the median read length and GC content accounted for only 0.31% and 2.75% of the R2 value in the full model respectively. However, we found that contig length, mean coverage and the simulated level of damage all accounted for approximately one third of the R2 value in our best performing model, indicating that these are the predictor variables of importance.

Table 1 The F1 score and Nagelkerke’s R2 mean values for the top ten models (ranked by F1).

The model we retained is highlighted in bold.

Variables	F1	R2	
readlength	0.791	0.001	
GCcontent/readlength	0.642	0.005	
damage/contiglength/coverage	0.624	0.607	
damage/contiglength/readlength/coverage	0.624	0.610	
damage/contiglength/GCcontent/readlength/coverage	0.623	0.619	
damage/contiglength/GCcontent/coverage	0.622	0.618	
damage/contiglength	0.600	0.432	
damage/contiglength/readlength	0.593	0.434	
damage/coverage	0.592	0.385	
damage/readlength/coverage	0.588	0.387	

Figure 2 Measures of model fit calculated on the test data for the top 3 models with one, two, three, four, and five variables, where red is the F1 score and blue is Nagelkerke’s R2.

Error bars indicate two standard deviations calculated from ten-fold cross validation.

Our final logistic regression model identified mean coverage, the level of damage, and the contig length as significant predictor variables for model accuracy. Each of these variables had positive coefficients, meaning that an increase in damage, genome coverage, or contig length all lead to improved model accuracy. Each variable contributed about one third weight to the R2 value in the model, indicating roughly equal importance in the accuracy of PyDamage. We integrated the best logistic regression model in PyDamage, with the StatsModels (Seabold & Perktold, 2010) implementation of GLM to provide an estimation of PyDamage ancient contig prediction accuracy given the amount of damage, coverage, and length for each reference (Fig. 3), and found these predictions to adequately match the observed model accuracy for our simulated data set (Fig. 4).

Figure 3 Predicted model accuracy of simulated data.

The grey title box above each panel is the simulated damage frequency on the 5′ end. Light blue indicates improved model accuracy, with parameter combinations resulting in better than 50% accuracy are outlined in green.

Figure 4 Observed model accuracy of simulated data.

The grey title box above each panel is the simulated damage frequency on the 5′ end. Light blue indicates improved model accuracy, with parameter combinations resulting in better than 50% accuracy are outlined with green lines. White tiles represent parameter combinations that were not sampled.

Application of PyDamage to archeological samples

To test PyDamage on empirical data, we assembled metagenomic data from the paleofeces sample ZSM028 with the metaSPAdes de novo assembler. We obtained a total of 359,807 contigs, with an N50 of 429 bp. Such assemblies, consisting of a large number of relatively short contigs, are typical for de novo assembled aDNA datasets (Wibowo et al., 2021). After removing sequences shorter than 1,000 bp, 17,103 contigs were left. PyDamage (revision 099fd34) was able to perform a damage estimation for 99.75% of these contigs(17,061 contigs). Because the ZSM028 sequencing library was not treated with uracil-DNA-glycosylase (Rohland et al., 2015), nor amplified with a damage suppressing DNA polymerase, we expect a relatively shallow DNA damage decay curve, and thus filtered for this using the pdj parameter. We chose a prediction accuracy threshold of 0.67 after locating the knee point on Fig. 5 with the kneedle method (Satopaa et al., 2011). After filtering PyDamage results with a q-value ≤0.05, pdj≤0.6, and prediction accuracy ≥0.67, 1,944 contigs remain. The 5′ damage for these contigs ranges from 4.0% to 45.1% with a mean of 14.3% (Fig. 6). Their coverage spans 6.1X to 1, 579.8X with a mean of 65.6X, while their length ranges from 1,002 bp to 90,306 bp with a mean of 5,212 bp and an N50 of 10,805 bp.

Figure 5 Number of ZSM028 contigs filtered by PyDamage with a q-value ≤0.05 as a function of the predicted prediction accuracy.

In total, 12,271 of the 17,061 contigs were assigned q-value ≤0.05. The red vertical line is the predicted accuracy threshold of 0.67.

Figure 6 Damage profile of PyDamage filtered contigs of ZSM028.

The center line is the mean, the shaded area is ± one standard-deviation around the mean.

The Kraken2 taxonomic profile of the microbial contigs identified by PyDamage identified as ancient (Fig. 7) is consistent with bacteria known to be members of the human gut microbiome, including Prevotella (239 contigs), Treponema (166 contigs), Bacteroides (103 contigs), Lachnospiraceae (119 contigs) Blautia (36 contigs), Ruminococcus (25 contigs), Phocaeicola (18 contigs) and Romboutsia (16 contigs) (Schnorr et al., 2016; Pasolli et al., 2019; Singh et al., 2017), as well as taxonomic groups known to be involved in initial decomposition, such as Clostridium (145 contigs) (Hyde et al., 2017; Harrison et al., 2020; Dash & Das, 2020). In addition, eukaryotic contigs were assigned to humans (18 contigs), and to the plant families Fabaceae (18 contigs) and Solanaceae (18 contigs), two families of economically important crops in the Americas that include beans, tomatoes, chile peppers, and tobacco. The remaining contigs were almost entirely assigned to higher taxonomic levels within the important gut microbiome phyla Bacteriodetes, Firmicutes, Proteobacteria, and Spirochaetes, as well as to the Streptophyta phylum of vascular plants. Collectively, these five phyla accounted for 1,283 of to 1,494 contigs that could be taxonomically assigned.

Figure 7 Taxonomic assignation by Kraken2 of the contigs filtered by PyDamage with q-value ≤0.05, pdj≤0.6, and prediction accuracy≥0.67.

Functional annotation of the authenticated ancient contigs using Prokka was successful for 1,901 of 1,944 contigs. Among these, multiple genes of functional interest were identified, including contigs annotated as encoding the multidrug resistance proteins MdtA, MdtB, and MdtC, which convey, among other functions, bile salt resistance (Nagakubo et al., 2002) (Table 2). Kraken2 taxonomic profiling of these three contigs yields a taxonomic assignation to the gut spirochaete Treponema succinifaciens, a species absent in the gut microbiome of industrialized populations, but which is found globally in societies practicing traditional forms of subsistence (Obregon-Tito et al., 2015; Schnorr et al., 2014). Other authenticated contigs contained genes associated with resistance to the natural antimicrobial compounds fosmidomycin, colistin, daunorubicin/doxorubicin, tetracycline, polymyxin, and linearmycin. A growing body of evidence supports an ancient origin for resistance to most classes of natural antibiotics (D’Costa et al., 2011; Warinner et al., 2014; Christaki, Marcou & Tofarides, 2020; Wibowo et al., 2021).

Table 2 Contigs assembled by metaSPAdes, identified by PyDamage as carrying damage, and annotated as carrying resistance genes by Prokka.

Contig name	Contig length (bp)	Coverage	Product	
NODE_2446	3232	64.3	Arsenical-resistance protein Acr3	
NODE_45	28638	26.0	Bifunctional polymyxin resistance protein ArnA	
NODE_832	6259	46.3	Cobalt-zinc-cadmium resistance protein CzcA	
NODE_832	6259	46.3	Cobalt-zinc-cadmium resistance protein CzcB	
NODE_2661	3058	91.5	Colistin resistance protein EmrA	
NODE_2661	3058	91.5	Colistin resistance protein EmrA	
NODE_215	13020	27.0	Daunorubicin/doxorubicin resistance ATP-binding protein DrrA	
NODE_136	16294	26.0	Daunorubicin/doxorubicin resistance ATP-binding protein DrrA	
NODE_1676	4090	81.3	Fosmidomycin resistance protein	
NODE_8410	1542	77.3	Linearmycin resistance ATP-binding protein LnrL	
NODE_29	35207	27.8	Multidrug resistance ABC transporter ATP-binding and permease protein	
NODE_232	12485	31.9	Multidrug resistance protein MdtA	
NODE_97	19553	27.4	Multidrug resistance protein MdtA	
NODE_12	45672	45.6	Multidrug resistance protein MdtA	
NODE_10	46280	59.8	Multidrug resistance protein MdtA	
NODE_97	19553	27.4	Multidrug resistance protein MdtB	
NODE_97	19553	27.4	Multidrug resistance protein MdtB	
NODE_12	45672	45.6	Multidrug resistance protein MdtC	
NODE_10	46280	59.8	Multidrug resistance protein MdtC	
NODE_232	12485	31.9	Multidrug resistance protein MdtC	
NODE_17	41269	29.9	Multidrug resistance protein MdtK	
NODE_465	8695	37.5	Tetracycline resistance protein TetO	
NODE_204	13262	44.9	Tetracycline resistance protein, class C	

Discussion

De novo sequence assembly is increasingly being applied to ancient metagenomic data in order to improve lower rank taxonomic assignment and to enable functional profiling of ancient bacterial communities. The ability to reconstruct reference-free ancient genes, gene complexes, or even genomes opens the door to exploring microbial evolutionary histories and past functional diversity that may be underrepresented or absent in present-day microbial communities. A critical step in reconstructing this past diversity, however, is being able to distinguish DNA of ancient and modern origin (Warinner et al., 2017). Characteristic forms of damage that accumulate in DNA over time, such as DNA fragmentation and cytosine deamination, are widely used to authenticate aDNA (Orlando et al., 2021) and have been important, for example, in enabling the reconstruction of the Neanderthal genome from skeletal remains contaminated with varying levels of modern human DNA (Briggs et al., 2007; Bokelmann et al., 2019; Peyrégne et al., 2019).

Nevertheless, applying such an approach to complex ancient microbial communities, such as archaeological microbiome samples or sediments, is more challenging. Existing microbial reference sequences in databases such as NCBI RefSeq have been found to be insufficiently representative of modern microbial diversity (Pasolli et al., 2019; Manara et al., 2019), let alone ancient diversity, making reference-free de novo assembly particularly desirable for both modern and ancient microbial metagenomics. However, de novo assembly of aDNA has always been a challenge due to its highly fragmented nature. While tools have been designed to improve the assembly of ancient metagenomics data (Seitz & Nieselt, 2017), assessing the damage carried by the assembled contigs has remained an open problem.

Existing tools such as HOPS (Hübler et al., 2019) and mapDamage2 (Jónsson et al., 2013) are readily available programs used to investigate ancient DNA deamination damage. However they perform a very different analysis compared to PyDamage, and their method, scope of application, and ability to scale are dissimilar to PyDamage. While HOPS applies additional authentication criteria beyond deamination (e.g., edit distance), when it comes to C → T substitutions, it uses a simpler heuristic to segregate damaged from non-damaged sequences, and does not provide any statistical testing of the damage. Furthermore, it is intended to be used in a targeted manner, with the user having to specify a list of known organisms of interest to look for, which is incompatible with de novo assembly that can potentially recover previously unknown species. Regarding mapDamage2, while it provides a statistical framework for aDNA damage modelling that can be used to rescale alignment quality score, it is not intended to be used in a metagenomics context. MapDamage2 does not provide a statistical test of the damage, and it is only designed to be used for single genomes, which poses scalability issues when using it with thousands of references typically generated by metagenomics de novo assembly.

Here, we have presented PyDamage as a tool to rapidly assess aDNA damage patterns for numerous reference sequences in parallel, allowing fast damage profiling of metagenome assembled contigs. To evaluate the performance of PyDamage model fitting and statistical testing, we benchmarked the tool using simulated assembly data of known coverage, length, GC content, read length, and damage level. Because PyDamage predicts based on C → T transition frequency, we originally expected GC-content to impact the available number of possible C → T transitions, and hence influence the predictions of PyDamage. However we found that GC content and read length were not a major driver of the accuracy of PyDamage’s predictions, but contig length, coverage, and damage level each played major roles. Taken together, this three parameter combination greatly influenced the ability of PyDamage to make a accurate damage assessments for a given contig. Overall, PyDamage has highly reliable damage prediction accuracy for contigs with high coverage, long lengths, and high damage, but the tool’s power to assess damage is reduced for lower coverage, shorter contigs length, and lower deamination damaged contigs. Although aDNA damage levels (cytosine deamination and fragmentation) are features of the DNA itself and out of the researcher’s control, we show that researchers can generally improve model accuracy through deeper sequencing.

When comparing the parameter range of our simulated data to real world de novo assembly data, we find that some of PyDamage prediction accuracy limitations are mitigated by the assembly process itself: de novo assemblers usually need a minimum of approximately 5X coverage to assemble contigs (Fig. 8) (Wibowo et al., 2021), and it is common practice to discard short contigs (<1000 bp) before further processing steps in a classical metagenomic de novo assembly analysis process. Nevertheless, low coverage, low damage, short contigs will remain a marginal challenge for damage characterization, even with further manual inspection. For example, for a 10,000 bp de novo assembled contig with 5% damage, PyDamage will only start to make reliable predictions once a coverage of 12X is reached (Fig. 3, interactive app available at https://maxibor.shinyapps.io/pydamageglm). For a similar contig with 10% damage, model accuracy is high even from 1X coverage. Overall, we find that PyDamage generally performs well on ancient metagenomic data with > 5% damage, but contig length and coverage are also essential factors in determining the model accuracy for a given contig.

Figure 8 Distribution of the coverage for ZSM028 contigs > 1,000 bp assembled by metaSPAdes.

Although we used the kneedle method (Satopaa et al., 2011) to select the prediction accuracy threshold for paleofeces sample ZSM028, users can adjust the selected prediction accuracy threshold according to the needs of their research question. For example, for some research questions where high accuracy in verifying damage is paramount, more stringent thresholds can be applied to minimize false positives, even though this increases false negatives. For other questions and where additional authentication criteria are available (such as taxonomic information or metagenomic bins), lower thresholds may be applied to reduce the number of false negatives due to insufficient coverage or contig length.

PyDamage is designed to estimate accumulated DNA damage in de novo assembled metagenomic sequences. However, although DNA damage can be used to authenticate DNA as ancient, it is important to note that it is not necessarily an indicator of intra vitam endogeneity. DNA within ancient remains typically consists of both an endogenous fraction present during life and an exogenous fraction accumulated after death. For skeletal remains, the endogenous fraction typically consists of host DNA, as well as possibly pathogen DNA if the host was infected at the time of death. For paleofeces or dental calculus, the endogenous fraction typically consists of microbiome DNA, as well as trace amounts of host, parasite, and dietary DNA. In both cases, the endogenous fraction of DNA is expected to carry DNA damage accumulated since the death (skeletal remains, dental calculus) or defecation (paleofeces) of the individual. Within the exogenous fraction, however, the DNA may span a range of ages. Nearly all ancient remains undergo some degree of degradation and decomposition, during which either endogenous (thanatomicrobiome) or exogenous (necrobiome) bacteria invade the remains and grow (Hyde et al., 2017; Harrison et al., 2020; Dash & Das, 2020). DNA from bacteria that participated early in this process(shortly after death or defecation), will carry similar levels of damage as the endogenous DNA because they are of similar age. In contrast, more recent necrobiome activity will carry progressively less age-related damage, and very recent sources of contamination from excavation, storage, curation, and laboratory handling are expected to carry little to no DNA damage.

To demonstrate the utility of PyDamage on ancient metagenomic data, we applied PyDamage to paleofeces ZSM028, a ca. 1,300-year-old specimen of feces from a dry rockshelter site in Mexico that was previously shown to have excellent preservation of endogenous gut microbiome DNA and low levels of environmental contamination (Borry et al., 2020). Using PyDamage, we assessed the damage profiles of contigs with lengths >1,000 bp, and authenticated nearly 2,000 contigs as carrying damage patterns consistent with ancient DNA. The overwhelming majority of these contigs were consistent with bacterial members of the human gut microbiome, as well as expected host and dietary components, but a small fraction of authenticated contigs were assigned to environmental bacteria and fungi, including the exogenous soil bacteria Clostridium botulinum (22 contigs) and Clostridium perfringens (38 contigs). These taxa are known to be important early decomposers in the necrobiome (Harrison et al., 2020), and the damage they carry suggests that they likely participated in the early degradation of the paleofeces before decomposition was arrested by the extreme aridity of the rockshelter.

Among the PyDamage authenticated contigs assigned to gut-associated taxa, NODE_10, NODE_12, and NODE_97 are of particular interest. These contigs encode a multidrug resistant ABC (MdtABC) transporter associated with bile salt resistance in the bacterium T. succinifaciens. T. succinifaciens is a human-associated gut species that is today only found in the gut microbiomes of individuals engaging in traditional forms of dietary subsistence (Obregon-Tito et al., 2015; Schnorr et al., 2014; Angelakis et al., 2019). It is not found in the gut microbiomes of members of industrialized societies, and is believed extinct in these groups (Schnorr et al., 2016). Its identification within paleofeces provides insights into the evolutionary history of this enigmatic microorganism and its functional adaptation to the human gut (Schnorr et al., 2019). The additional identification of other resistance genes among the authenticated contigs provides further evidence regarding the evolution of antimicrobial resistance in human-associated microbes.

Conclusion

As the fields of microbiology and evolutionary biology increasingly turn to the archaeological record to investigate the rich and dynamic evolutionary history of ancient microbial communities, it has become vital to develop tools for assembling and authenticating ancient metagenomic DNA. Coupled with aDNA de novo assembly, PyDamage opens up new doors to explore and understand the functional diversity of ancient metagenomes.

We thank Nigel Bean and Jonathon Tuke for extremely useful discussions.

Additional Information and Declarations

Competing Interests

Author Contributions

Data Availability

The authors declare there are no competing interests.

Maxime Borry, Alexander Hübner and Adam B. Rohrlach conceived and designed the experiments, performed the experiments, analyzed the data, prepared figures and/or tables, authored or reviewed drafts of the paper, and approved the final draft.

Christina Warinner analyzed the data, authored or reviewed drafts of the paper, and approved the final draft.

The following information was supplied regarding data availability:

The genetic data for ZSM028 is available on the European Nucleotide Archive (ENA): PRJEB33577.

The PyDamage Software and source code available from Github: https://github.com/maxibor/pydamage, license: GPLv3.

The code to replicate the simulation of reads and contigs, and the figures is available at GitHub: DOI: https://doi.org/10.5281/zenodo.4981768.

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
