# Peer review of "PyDamage: automated ancient damage identification and estimation for contigs in ancient DNA de novo assembly"

_PeerJ, doi:10.7717/peerj.11845_

## Round 0.1 · original submission · Minor Revisions

Thank you for submitting your work to PeerJ. Apologies for the time it took to get these reviews. For some reason, I find that nowadays reviewers are hard to find, as everyone seems to be really busy. Your manuscript was evaluated by two reviewers. I am glad to see that both reviewers made constructive criticisms of your work and, although Reviewer 1 suggested your manuscript requires Major Revisions, I am sure you will not have any problems addressing his/her comments to his/her satisfaction. Please read their reviews carefully and incorporate/address the changes requested, especially related to the meaning of the written sentences. Also, make sure your code and test data are available in public repositories. I am looking forward to receiving a revised manuscript.

·

Basic reporting

The manuscript by Borry et al. is well presented. It is written in straightforward and clear English. The authors introduce the manuscript with ample background information and context so the non-expert reader can follow what is going on.

The paper is coherent and well-structured in the conventional and unsurprising article format, and figures and tables are beautiful and mostly easy to understand and relevant. The working hypotheses, while implicit, are clear enough. Overall, I applaud the authors for writing such an easily-understandable article on a technical topic.

There is plenty of relevant references with few exceptions

Experimental design

Generally, the central research question is well defined, meaningful and fill a clear and explicitly stated gap in knowledge. Technically, the paper is fairly clear in its intentions and methods, and described in detail enough to replicate. Code and data is publicly and easily available.

Comments:
Line 148: It is not clear why the lambda value approaches a chi-square distribution with two parameters. If this is well-known, a reference would be needed. If this can be deduced mathematically, please include a proof or explanation.

Line 169: In the model selection, the use of Nagelkerke’s R-squared raises some issues. Its concrete value is difficult to interpret, and it’s not clear why it is preferred over simpler measures like receiver operating characteristic (ROC) area under curve (AUC), F1-statistic, or similar. In particular, given that Table 1 shows that the selected model is inferior according to the balanced accuracy – a measure which is much more straight forward and presumably more relevant, the discrepancy between R-squared and accuracy should at least be investigated and discussed.

Furthermore, there is no discussion about recall/precision tradeoff. Any choice here would be arbitrary, and many choices could be defended, but it is still relevant. For example, suppose different models could tradeoff recall/precision at different rates between recall and precision. Then different tradeoffs would favor different models to appear superior. The authors could address this by either deciding balanced accuracy was the relevant criterium and fit the models to optimize this instead of sum of squared error, or by showing AOC curves or similar for each model to explicitly show the tradeoff.

Validity of the findings

Underlying data has been provided, and reproducibility is greatly improved by a code repository containing all relevant code.
Overall, the conclusions are well supported, with some caveats.

Comments:
Lines: 209-221: While the contigs assigned as ancient were analyzed, there is no mention of the remaining contigs. According to the implicit hypothesis, these should presumably be from clades associated with soil metagenomes, and not gut metagenomes, but this is not investigated. A simple comparison between the ancient versus modern contigs in order to show that the populations are clearly different would greatly strengthen the author’s argument.

Lines 254-256: It is not clear from the article that PyDamage is “highly reliable”, in the sense of having a near-100% accuracy. The only place where I can find direct measures of accuracy is Figure 4, from which it is very difficult to tell e.g. 95% accuracy from 99% accuracy. If this statement should be included in the conclusion, there should be at least one place in the article where the authors directly measure accuracy and show it in a clear figure or in text.

Lines 273-275: This is not clear from the figure. If the figure is cited for this claim, I would re-make figures 3 and 4 in a way where it is possible to read this more precisely.

Figures 3-4: Besides the general comments above, it is not clear from either the figure nor the legend that the numbers above each of the 10 boxes represent damage frequency. Please add this to the legend or the figure itself.

Additional comments

Lines 48-50: Is this supported by a reference? If so, please add it. If not, please explicitly state it is a hypothesis.

Lies: 94-96: For clarity, and if word count allows it, consider mentioning explicitly why GC content could be an interesting variable for PyDamage. Considering that the conclusion explicitly mentions that it turns out to not be important, it would be nice it you explicitly stated why you investigated it. Perhaps something like “because PyDamage predicts based on C->T transition frequency, we expect GC-content to impact the available number of possible C->T transitions, and hence could influence the predictions of PyDamage”.

Line: 198: Specify that “filter for” means retain, not remove. This is ambiguous in English. In this particular case, it could mean either retain long contigs (because PyDamage works better on long contigs), or retain small contigs (because you expect damaged ancient DNA to result in short contigs).

Lines 199-200: It is not clear what you mean by successful estimation. Do you mean that the program worked? Or that it was correct? And how do you know it was correct?

Reviewer 2 ·

Basic reporting

Basic reporting adheres to journal requirements.

Experimental design

No comment

Validity of the findings

No comment

Additional comments

This paper by Borry et al. tackles a really key issue in ancient metagenomics, and as such is clearly of very high interest to the community at large. As the authors state, there are a number of existing software that attempt to tackle this type of issue, but none that do this terribly well, particularly on meta-genome de novo assemblies. I was really impressed with the software, and the paper is written well. I don't have any major critiques, but have listed a few minor questions/comments below:


Line 129: 'non-default k-mer lengths...' How were these chosen?
Lines 172-173: This is a bit of an awkward sentence
Table 1: I can't seem to find where you give a justification of retained model choice.
Figure 3: The values in grey boxes need to be mentioned in the legend.
Lines 262-263: I'd be interested in more delimitation of the parameters affecting prediction accurracy. Some of this can of course be pulled from the figures, but it would be nice if these were explicitly stated.

Lines 76-77: You mention in the introduction some of the existing software, and some of the issues associated with them. Particularly HOPS and mapdamage2. I'd be interested to see some comparison of the sensitivity/scaleability issues you mention.

---

## Round 0.2 · accepted · Accept

Fantastic work. Thank you for submitting it to PeerJ.